# An Investigation into the Physical Activity Experiences of People Living with and beyond Cancer during the COVID-19 Pandemic

**DOI:** 10.3390/ijerph19052945

**Published:** 2022-03-03

**Authors:** Andy Pringle, Nicky Kime, Stephen Zwolinsky, Zoe Rutherford, Clare M. P. Roscoe

**Affiliations:** 1Human Sciences Research Centre, University of Derby, Kedleston Road, Derby DE22 1GB, UK; c.roscoe@derby.ac.uk; 2Bradford Institute for Health Research, Temple Bank House, Bradford Royal Infirmary, Bradford BD9 6RJ, UK; nicola.kime@bthft.nhs.uk; 3West Yorkshire & Harrogate Cancer Alliance, White Rose House, West Parade, Wakefield WF1 1LT, UK; s.zwolinsky@nhs.net; 4School of Public Health, Faculty of Medicine, The University of Queensland, Brisbane, QLD 4006, Australia; z.rutherford@uq.edu.au

**Keywords:** physical activity, football, intervention, cancer, COVID-19, football community trusts, implementation, qualitative investigation

## Abstract

This study investigated the physical activity experiences of people living with and beyond cancer (PLWBC) during the COVID-19 pandemic. Participants attended the cancer and rehabilitation exercise (CARE) programme delivered by a football community trust. Staff (*n* = 2) and participants (*n* = 9) attended semi-structured interviews investigating the PA participation and experiences of attending/delivering different modes of CARE, including exercise classes delivered outdoors and delivered online. Interviews also investigated participant aspirations for returning to CARE sessions delivered in person indoors. The findings show that the COVID-19 pandemic and government restrictions impacted on PA participation, yet exercise sessions provided via CARE offered participants an important opportunity to arrest their inactivity, keep active and maintain their fitness and functionality. Barriers to participation of CARE online included access to IT infrastructure, internet connectivity and IT skills and comfort using IT. Regarding CARE outdoors, the weather, range of equipment, variety of exercises and the lack of toilets and seats were barriers. In the different CARE modes, the skills of delivery staff who were sensitive to the needs of participants, social support, and the need for participants to maintain good mental and social health were important facilitators for engagement and are considerations for programme delivery. CARE helped PLWBC to keep physically active.

## 1. Introduction

The UK Chief Medical Officer highlights the beneficial role that physical activity (PA) can have for the prevention and management of several long-term conditions, including cardio-vascular disease (CVD), mental health and some cancers [1,2]. However, physical inactivity continues to be a public health concern; this is especially the case in those people who suffer with long-term conditions [1,2,3,4]. The 2016 Lancet report on PA identifies the need for collective action and innovative approaches adopted by a diverse range of sectors and organisations to encourage people to be physically active [5]. There has been an increase in the delivery of health improvement interventions offered through football community trusts, the charitable arm of professional football clubs, including those aimed at a range of long-term conditions and health enhancing behaviours [6,7]. Trusts deploy their unique reach of the ‘football brand’ to include people, product, processes, and places to connect and engage a range of PA and health priority groups [6,7,8]. A recent review of health improvement interventions delivered in the English Football League identified that all football community trusts provided PA interventions as well as programmes for mental, social and physical health and wellbeing [6]. Elsewhere, interventions have been offered by community trusts to local participants for tackling overweight and obesity [9], CVD [10], mental wellbeing [11], as well as health programmes aimed at children and young people [12], adults and older adults [11,12,13,14,15]. A recent scoping review of programme characteristics in community-based exercise programmes for people living with and beyond cancer (PLWBC) [16], identified interventions deploying football for the management of cancer, including prostate cancer [17,18]; however, no exercise programmes delivered by UK football club community foundations or trusts and aimed at supporting the PA participation of PLWBC were identified at that time [16]. More recently, Rutherford et al. investigated the impact and implementation of the Cancer and Rehabilitation Exercise (CARE) programme delivered face-to-face by Notts County Foundation (NCF) [19]. They concluded that a football community trust, delivered PA cancer intervention, was successful in significantly improving participants’ quality of life and in regaining the physical and psychological functioning of PLWBC.

Rutherford’s study [19] took place before the advent of COVID-19, in 2020, which will be known for the emergence of the global pandemic caused by the spread of the COVID-19 virus [20]. In the UK on 23 March 2020, unprecedented legal restrictions were imposed by the UK Government to prevent the spread of the virus [21]. The government restrictions led to new laws for how people lived their lives by instructing people to stay at home for all, but essential reasons, including essential work, food shopping, caring for vulnerable people, accessing health and social care and to undertake PA [21,22]. People were normally permitted to leave their home once per day to undertake PA [22]. However, restrictions also extended to the access of PA opportunities with non-essential services such as gyms, sports centres and health clubs forced to close their doors during the periods of government restrictions. The restrictions in the sport and leisure sector led to a temporary cessation of ‘face-to-face’ sport and recreation services delivered indoors, meaning that for some people, their PA routines and their networks which support their health were negatively impacted. Moreover, restrictions on leaving home for all but essential reasons, including work, meant that for some people, PA undertaken through paid or voluntary employment as well as commuting to their place of work or study was disrupted.

Further, it was recommended that ‘people who were identified as being at high risk of severe illness from COVID-19, follow more stringent restrictions compared to people without a high-risk status; high risk status was based on an existing chronic health condition such as coronary heart disease, chronic kidney disease and diabetes, aged 70 years or above and if a person was pregnant’ [23]. People considered as ‘clinically extremely vulnerable’ and especially suspectable to the contraction of the COVID-19 virus, also included some PLWBC and it has been suggested that based on available data, COVID-19 appears to affect people of all ages; however, those who are older and have pre-existing medical conditions, including cancer, may be at higher risk for serious medical complications [24]. PA participation is recommended throughout cancer therapy and survivorship [19], but the COVID-19 pandemic is expected to negatively impact on PA levels of PLWBC [25]. Moreover, it has also been suggested that non-face-to-face community support is urgently needed for adults facing reduced levels of PA and psychological hardships due to the COVID-19 pandemic [26].

The emergence of the COVID-19 pandemic and subsequent government restrictions has created both new challenges, as well as opportunities for PA. The outbreak of the pandemic has accelerated the use of modern technology in the fitness sector [27]. For instance, there has been emerging opportunities to be physically active using online digital platforms with PA sessions being delivered on software applications such as Microsoft Teams and Zoom [28]. PA has also been promoted via other digital platforms in the form of exercise videos and exercise information sheets detailing the frequency, intensity, time, and type of exercises that can be performed. These measures being made available by exercise provider websites and through email communications to the service users and contacts held by host organisations. When legally permitted, PA sessions have also been provided in outdoor spaces such as parks and countryside. Adapted PA provision also extends to the offer made by charitable organisations, such as Football Club Community Foundations and Trusts, to help people keep active during the COVID-19 pandemic and during periods of government restrictions when indoor face-to-face services were not permissible.

The National Cancer Research Institute has identified research priorities which, in part, aim to establish the ‘best’ ways to support PLWBC to make lifestyle changes to improve their health’ [29]. Further, few studies have investigated the PA experiences of people attending PA interventions aimed at the management of cancer and specifically delivered through community football foundations [19], including those delivered during the COVID-19 pandemic and through periods of government restrictions. This issue is especially contemporary as COVID-19 government restrictions have been re-initiated in the UK and in parts of Europe in the Winter of 2021–2022 [30]. Investigations into the PA experiences of PLWBC during these periods provide an important opportunity to reflect on how this group can be supported to keep physically active. Further, it has been suggested that as the field of PA for cancer survivorship aims to move research into practice, there is an opportunity to learn from practice-based evidence emerging from real-world programmes [16].

This is one of the first studies to investigate the PA experiences of PLWBC who engaged in a cancer and exercise rehabilitation programme offered by a Football Club Community Trust during the COVID-19 pandemic. Participants engaged CARE, using different modes of delivery (i.e., exercise sheets, pre-recorded videos, exercise outdoors and the Zoom platform). Our study also investigates the experiences of the staff who delivered and supported these modes of CARE delivery. Finally, we explore participant aspirations for returning to CARE sessions delivered face-to-face at indoor venues following the easing of the COVID-19 restrictions.

## 2. Materials and Methods

### 2.1. Intervention Context

The CARE programme is delivered by NCF, an established football community trust organisation with a proven track record in delivering successful community-based health improvement interventions [10,19]. As Rutherford and colleagues report, in response to local needs, ‘NCF, in partnership with Macmillan Cancer Support and informed by the Macmillan Physical Activity Behaviour Change Care Pathway; designed, funded, and established CARE in March 2015, for PLWBC across Nottinghamshire, England. Service users were referred onto CARE via various health care providers (e.g., Cancer Nurse Specialists, General Practitioners, Occupational Health Consultants, Oncologists, Physiotherapists, Practice Nurses, Psychologists, Radiographers), or via self-referral. The CARE programme was overseen by an appropriately qualified (L4 Cancer and Exercise Rehabilitation https://pdphub.com/event-pro/cancer-rehabilitation-foundation-training/ (accessed on 27 October 2019) programme coordinator and delivered from the Portland Centre; a NCF run leisure centre situated in the City of Nottingham, as well as other parts of Nottinghamshire’ [19]. The impact of the CARE programme delivered through traditional face-to-face methods and at indoor venues has been reported elsewhere [19]. With the emergence of the COVID-19 pandemic, the implementation of COVID-19 restrictions [23] and the closure of all but essential services, NCF were required to pause the delivery of face-to-face sport and leisure services including those held at the Portland Centre. However, in aspiring to meet the needs of CARE participants, a decision was made by NCF to offer the CARE programme to participants using the following modes of delivery as and when permitted under the government COVID-19 restrictions in place at the time.

#### 2.1.1. CARE Exercise Sheets

CARE instructors prepared exercise sheets detailing the different exercises that participants typically performed during the CARE sessions. These were sent out to CARE participants via their Microsoft outlook email. Participants could download the information sheets, print off and undertake the exercises in their own home at their convenience.

#### 2.1.2. CARE Pre-Recorded Exercise Videos

CARE instructors filmed example exercise sessions, and these were housed on a video platform and participants were sent the link in order that they could access the videos on their electronic devices and undertake the exercises in their own home at their convenience.

#### 2.1.3. CARE Sessions Delivered Online via the Zoom Platform

The CARE sessions were delivered by the instructors and offered live on the Zoom platform and accessed by participants in their own homes using their PC, laptop, smart phone, or another Wi-Fi-connected device. Sessions took place up to three times a week, lasting approximately 60 min in total and were delivered by appropriately qualified instructors. The instructors built in social time for participants to chat to each other after the session using the breakout room function on the online software. The social aspect was a feature of the face-to-face CARE classes delivered pre-pandemic [19].

#### 2.1.4. CARE Sessions Delivered Outdoors in Green Space

When the UK Government COVID-19 restrictions were relaxed and outdoor group exercise sessions were legally permitted, instructor-led CARE sessions were delivered live in person outside in a local green space close to the Portland Centre known as the ‘Embankment’. Instructors took equipment that was portable such as dumbbells, exercise bands, mats and the natural built environment was also harnessed, e.g., steps for step-related exercises. The exercise sessions were run up to twice times per week for a duration of approximately an hour. Participants and staff observed the COVID-19 guidance issued at that time and supported that the risk of transmission of the virus was reduced due to outdoor ventilation, and this is important for vulnerable groups. Prior to this, a group of CARE participants self-organised their own exercise session when permitted within the COVID-19 guidance at the time. Participants took exercise equipment and music and followed the CARE exercise routines and sessions ran twice per-week.

#### 2.1.5. Self-Organised Physical Activity

In addition to the more structured CARE sessions described above, outside of the CARE offer, participants were encouraged to undertake their own self-directed PA. This could be performed on their own or with others indoors and/or outdoors, as, and when permitted under the COVID-19 government restrictions that were in force at that specific time. Therefore, CARE participants had several options to maintain their PA participation during the COVID-19 pandemic through CARE and self-directed PA.

### 2.2. Participant Recruitment and Ethical Procedures

Ethical approval was obtained through the University of Derby, College of Science and Engineering Research Ethics Committee (ID: ETH 2021/1038) in January 2021. Prior to any data collection, all participants were provided with a participant information letter, a consent form, and all participants provided written consent to take part in this research. Participants in this research (CARE service users and delivery staff) received an email invitation to participate in the research and volunteers were requested to contact members of the research team either by email or phone to indicate their intention to take part in the research. Following this request, the researchers then contacted the participants directly using the details they provided to confirm their participation and arrange an interview, by either Microsoft Teams or by phone. Delivery staff who took part in the interviews were contacted directly via email and invited to participate in an interview by the research team [19]. Participants were asked for permission to record the interview using a digital voice recorder and where permission was not granted, the researchers asked for permission to make notes during the interview.

### 2.3. Interviews with Participants

Interviews took place between May and July 2021. Data were collected using a semi-structured interview lasting approximately 45 min and which aimed to identify participant profile age, gender, type of cancer, PA participation, including the mode by which people took part in CARE. This included what factors worked well, as well as aspects of the delivery that could be improved. Nine CARE participants and two CARE delivery staff members elected to take part in the research, and this reflects approximately one third of participants who attended CARE sessions before the pandemic. The interview design and schedule were adapted from the approach used by Rutherford et al. [19] and supplemented with additional questions to reflect the scope of the research, including the delivery of sessions outdoors and online. The interview schedule is available by contacting the corresponding author Prior to data collection, all procedures were piloted and reviewed for acceptability and accessibility in line with previous approaches [6]. The researchers took a reflexive approach to review how the interviews had gone on a regular basis [6,19].

### 2.4. Data Analysis

Following the participant and staff interviews, the recorded audio files were transcribed verbatim by a professional transcribing service. Braun and Clarke’s six stages of thematic analysis were deployed, following transcription and immersion of the interview transcripts to saturation, coding identified interesting features in the data, and these were grouped into coherent themes to provide an analytical framework [19,31]. A visual map was developed by hand to show the themes and their relationship. Two researchers (A.P. and N.K.) met to refine the specifics of the themes and to generate clear definitions and names for them, which were then shared with the other authors. This approach has commonly been used in the investigation of football-led health improvement programmes [6,19]. Five main themes were identified: referral to the CARE programme, PA participation (including engagement in the different CARE modes), barriers to PA participation via CARE, facilitators for PA participation via CARE and feelings about the future participation in CARE sessions following the easing of the COVID-19 restrictions.

## 3. Results

### 3.1. Demographics and Participant Profile

Table 1 shows demographics, cancer types, modes of CARE and self organised PA that participants engaged in.

### 3.2. Referral to the CARE Programme

Participants were recruited to CARE through several pathways and reflect a range of ages.

People were recruited in a variety of ways. So, we’ve got our specialists and consultants, oncologist nurses that they have obviously contact with at their appointments in hospitals. Support groups that are around, so we make use of Maggie’s Centre, we’re in their quite often doing presentations and things. Then it’s also self-referral, so from seeing posters around different areas. (Staff 2)It was the Late Effects Team that referred me to the physio and then it was the physio who mentioned about CARE. So that’s how I was able to access it really. (CARE 4)Our youngest participant is in their early 20s and our oldest is in towards their late-80s, so you can see a wide variety. However, the bulk of our participants have probably been in that 50 to 70 range, that’s the average age I would say, although it is open to everyone, and we get obviously a real variety of people. (Staff 1)

### 3.3. Physical Activity Participation

#### 3.3.1. Physical Activity Participation before Cancer Diagnosis

Participants reported being physically active or inactive before their cancer diagnosis and those that were participating in regular exercise engaged in several different modes of activity; this is illustrated by the following quotes.

I was fit and well before diagnosis, but I was never one for going to the gym and so I needed a lot of information when I joined CARE. (CARE 5)[I had] a soft tissue sarcoma in my leg. And I went on to have surgery and quite lengthy six-week daily radiotherapy. And at the time I was reasonably active because I’d been recently retired. And I, you know, did a lot of Pilates. I did a lot of walking. I went swimming a bit, I didn’t like swimming that much, but I went because I thought it was good for me. (CARE 8)My general health wasn’t very good anyway. I suffer a lot with depression and anxiety and the GP referred me to what was called the Be Active scheme, which is run by various gyms, so I went there. They put you under basically a personal trainer there who guide you through various things to help you cope with depression and anxiety because obviously exercise is supposed to help it. When I was 20, I had cancer before, so I’ve never really done physically at all, I was never a sporty person. (CARE 9)We kind of split them [participants] up into two groups, it’s one extreme or to the other. We never have someone who’s quite physically active or here and there, it’s always we have the participant who is, I cycle five times a week and I do 80 miles per week, or, I don’t do anything, it’s always one or the other. (Staff 1)

#### 3.3.2. Physical Activity Participation Pre-COVID-19

Before the COVID-19 pandemic, most of the participants reported that they were physically active, had been engaging in the face-to-face CARE sessions delivered indoors and were undertaking self-directed PA.

I’m trying to do a sort of, an eclectic mix of stuff [activity], because I tend to think, “Oh well this is great”. And then I get bored with it. So, I dog walk every day obviously because I must. I’m still running outside now that I’m allowed to go outside, I am running. I’ve just started swimming in the lake. (CARE 7)

#### 3.3.3. Physical Activity Participation during COVID-19

CARE participants reported different preferences for engaging in the CARE programme.

We decided we had a variety of different ways to interact with them and they can pick and choose what’s going to be better or necessary, whether that be Live Zoom classes which were well-attended but not as well-attended as the normal face to face sessions. (Staff 2)It was probably up to three quarters of CARE attendees did Zoom, so a quarter of them were saying they don’t want to do anything because they didn’t want to do Zoom, they didn’t want to do any PA or rehabilitation, some wanted to do [CARE] rehabilitation in person and some wanted to do [CARE] on Zoom. So, we had these three sets of groups who were happy with what they wanted and tried to form them into the same group online. (Staff 1)

#### 3.3.4. No Physical Activity

In some cases, participants reported doing no PA.

I have done nothing. I have back problem and I have physio, but if I had been going to CARE, I would not have this, I have gone backwards since CARE ended in person. (CARE 5)

### 3.4. Physical Activity via CARE

#### 3.4.1. Physical Activity via CARE on the Zoom Platform

Staff and participants reported on CARE delivered via Zoom.

I would say 75% of our overall participants engaged in some sort of online activity, so the Zoom classes probably were quite a good hit, but again we’re normally averaging 20/25 numbers and we’re probably getting 10 to 15 tops for the Zooms. (Staff 2)I did not attend Zoom CARE, I am not happy with Zoom on my phone, I did not enjoy it and I did not have the facility to do Zoom on a big TV, I have done absolutely zilch since CARE ended. (CARE 5)I have never got back the mobility that I had before cancer. And it’s got a lot worse in lockdown, probably understandably. I wasn’t even going for even the permitted walks because I was sitting round all day. And everything has really ceased up now. So, when CARE came back with the Zoom sessions I thought, well this is it now, you know, this will make me do it, this will be good. (CARE 8)When they put the [CARE] online up, I thought, yes…I was doing Pilates Tuesday morning and two CARE sessions on Zoom as well. So, it meant that I was getting all my usual exercise in which was fabulous. It’s been fabulous, just to be able to keep going with the CARE. You know, we were thrilled. I mean, some people didn’t do it because they don’t like using Zoom. That didn’t bother me at all because I say, I’m very adaptable…And my motto is, where there’s a will, there’s a way. Nothing is impossible. (CARE 2)I’m not good on Zoom. I can do it one-to-one or with a couple of people. But when you’ve got more, I find I really struggle with it. I used to struggle when I had to do it for work a few years ago. And even with friends, if there are more than three or four, I just find the whole experience a bit intimidating. I suppose a bit of it is trying to work out who is going to talk first, and I find that if you’ve got eight or nine people on a screen, I find the whole process a bit daunting, and it puts me off. And I find concentrating on the screen for more than about 20 min, it’s easier to talk to somebody on the phone or face to face with a cup of tea. (CARE 3)I just find the Zoom very, very easy. I mean, the instructors, he is very good at keeping us informed. Sends us a link every week. “This is the link for”, whatever and you just click on it and it’s absolutely fine. I think we’ve only really had one or two glitches and that was because of internet connections really. So, it’s not really been an issue. And I felt very comfortable with it. (CARE 4)

#### 3.4.2. Physical Activity via CARE Outdoors in Green Spaces

As with the CARE sessions online, participants also expressed different preferences for CARE sessions delivered outdoors.

During lockdown six of us, decided to carry on running the CARE and carried on doing it legally while we could. So, we started meeting by the Trent, so set up some exercise sheets, took some music down and cones…We took weights down and did normal weight exercises. We did, there are steps down there so we could do step exercises. We were using either the grass when it was dry with a mat on or a tree for press ups. We have typically nine or 10 stations and do that twice for an hour. (CARE 6)Well, I have been going to the CARE sessions outside on the Embankment. I mean it was great last autumn, you know, beautiful colours. And then in the spring of the blossom and blue sky. And it was just such a wonderful feeling. (CARE 8)I could not entertain the CARE park sessions, it is cold, there are no seats to sit [on], I don’t do cold weather and there are no toilets. There is not the variety of equipment, there is just the equipment that can be carried in the bag. (CARE 5)Yes, in fact that’s a bit of a downer for me as well because obviously it’s great that things are opening for everybody but since they’ve opened and started outdoor [CARE sessions], which I don’t really see any point in me going to really because I liked the fact that it was indoors. I liked it was because I could use the equipment before and after, plus after the session we had some time to play badminton and table tennis which I loved, so going out-doing it outdoors is just basically what I was doing on Zoom but in person, so it seemed a bit pointless for me, especially with the extra cost of getting there. (CARE 9)I haven’t been along to the outdoor sessions. I’ve got problems with mobility now. So, I can’t walk very much and I can’t drive very far. And I haven’t wanted to use public transport. (CARE 3)

#### 3.4.3. Physical Activity via CARE Videos and Exercise Sheets

Exercise videos and documents were sent out on a weekly basis so they can exercise from home and do their own. We had some great little exercise videos that they could follow which was brilliant and demonstrated them all through and they really seemed to enjoy those, but could we set little challenges or things that would be a bit more engaging, a little bit more fun, rather than just the same structure of, oh, it’s another video, which was great. (Staff 2)

#### 3.4.4. Self-Organised Physical Activity

Several participants engaged in a combination of CARE plus their own self-directed PA.

I am doing Yoga online. Yes, but it’s very- It’s quite low intensity yoga. It’s all stretching rather than strenuous moves. (CARE 9)In our village chair-based exercise group. The instructor left copies of all the music with us and we wrote down the exercises. So, I got the music out…and just stared doing sort of 10, 15 min in the afternoon, I try to do some most days just to- but they’re all chair-based ones and on the list, I’ve got it says whether it’s for your core, for your arms, for your legs, whatever. (CARE 1)

### 3.5. Barriers to Physical Activity Participation via CARE

#### 3.5.1. Attending CARE Online via the Zoom Platform

Access to IT infrastructure and/or IT hardware was a barrier when attending CARE via the online platform.

You need to know why people do not attend online through Zoom, we did not have the facilities and a camera that we could use. (CARE 5)And there was bandwidth, it kept dropping out or freezing. And in fact, I changed my broadband provider and I’ve also, upped some of my broadband allowance. I seemed to have different problems with Zoom every time I went into it. And I kept thinking that I don’t know why this is happening. It worked all right before. (CARE 8)Participants did join with phones. It might be low internet connection as well, on the mobile it’s the reception that they had, the internet speed that they had, so we’ll sometimes we’ll have people who might start an hour but also while might be breaking off and leaving the room as well at the same time but, yes, we just try and make it as comfortable as possible. (Staff 1)All this latest modern stuff, this Zooming and all that—I could cope with emails. I can cope with my mobile phone which is an old one and I can do texts on, but not a lot more than that. So, I didn’t getinvolved…I just have a thing about seeing people on the screen and that sort of thing. It doesn’t seem real to me. Well, no, it seems odd. (CARE 1).

The sizes of the exercise groups and the IT etiquette and processes of engaging CARE online were identified as barriers.

We did organise I think about four or five Zoom calls which was just a chance of team coffee as such over Zoom, and they worked quite well. A little bit difficult at times, again everyone’s trying to talk over everybody, but we use the breakout rooms and stuff and that was led by participants, it was one of their ideas and they led that and things which was great. (Staff 2)The breakout groups, we did try at one stage to have a sort of, you know, a social thing online. And then have a big group all together and then have breakout groups, that was the very beginning, you know, at the first lockdown. But that didn’t really work. But I think there’s an ideal group size on Zoom that makes you feel not so you’re sort of one of the five, and you’re feeling we have to go next time because there’s only five of them, and if I go, I don’t go and I know somebody else isn’t going for that time, it’ll only be three. (CARE 8)The troubles that we had were to do with it’s just that communication because if you have 10, 15 or 20 people in the same room and that’s their only chance to see other people on the CARE programme everybody wants to talk and you all have to wait to listen just for one person to talk, and then if someone’s going to respond you’re going to get 10 other people who are going to try and respond at the same time, so that communication. (Staff 1)

The capacity of the delivery staff was identified as a issue, but this did not stop taking part.

It was a prescribed time from the coach’s point of view, some were better than others at making you feel more relaxed. But some it was just like feeling we’re just getting through it. They [the instructors] were not watching you in detail, I mean with the fixed screen, I was saying, I’ll do this on the wall. But sometimes they’d started, and I hadn’t set myself up in time. So I think that was the negatives. (CARE 8)I think we could do more to try and engage with others, maybe change the style of the session on Zoom. I know it’s obviously difficult you’re limited to what you can do but look into different things we could do to make it a little bit more full and engaging rather than so repetitive. It was pretty much like a bootcamp class online as such so they’re copying movements or exercises we’re doing we’re watching them do it, can we incorporate different things within that, be a better way of doing it. (Staff 2)

The missing social element of online CARE sessions was identified as a barrier.

Before we went into lockdown, before the Zoom gyms, that’s what I call it, people did used to meet in the café afterwards [after the CARE sessions]. (CARE 4)Another thing that was missing [compared to the in person sessions], we always went for a drink- at the pub afterwards near the gym. Which was fantastic. But of course, that’s all gone as well with Zoom. Which is how you got to know people better. It’s just- It just feels different not to be face to face. (CARE 8)

The lack of confidentiality of conversations between staff and participants via the CARE online platforms.

You always felt like you were never in a safe space to have that one-to-one talk with someone [attendee] if you wanted to have that talk, as you usually would in a face-to-face session. (Staff 1)

Creating momentum and demand for the CARE online sessions was a barrier

I’d say the negative of Zoom was trying to adjust to get people motivated to do Zoom sessions, but also at the same time is when it snowballs it kind of picks up quite a lot but it’s just that starting process of changing the whole dynamic of the CARE programme. The Zoom sessions, it was very much limited numbers and then when it came to I’d say November time when we went into lockdown two so it started to funnel down quite a lot of just quite double digit in participants but it started to filter down to the most regular people who attend. (Staff 1)

#### 3.5.2. Attending CARE Sessions Delivered Outdoors in Green Spaces

A lack of facilities and the uneven ground at the outdoor venues were barriers that were reported.

I didn’t feel I could cope with going to the outdoor one when they started that up because I need to be near a toilet. And the Trent hasn’t really got any toilets there that you could use. (CARE 1).There are seats to set yourself when you exercise and a lack of exercise equipment. (CARE 5)I don’t balance very well on lumpy grass, fell, and twisted, well not fell, I twisted my ankle doing the outside sessions. Took two weeks off. Went back. And trying to be clever thinking, ‘I’m really going to do this’ you know, ‘I’m going to be good at this. I’m going to do this one without my stick’. So, I did. Skidded on the gravel (CARE 8)

Mobility issues and ability of participants to commute to the exercise venue were also reported as barriers.

Driving more than a couple of miles is really difficult. And when I was going to CARE regularly, I used to normally use public transport simply because I find driving, well, not so much driving, but parking round there, very stressful. So, I used to end up going into the sessions feeling really exhausted because I’d had a really stressful experience trying to park which was not a good start to the session. So, whenever possible, I used to use public transport. And I haven’t wanted to use public transport, though I am using it now, during the last year particularly. The other thing is I can only walk for about 15, 20 min at the moment. (CARE 3)

### 3.6. Facilitators for Physical Activity Participation Undertaken at CARE Sessions

Participants and staff reported the factors that helped participants with their engagement in PA, including participation at the CARE sessions undertaken online or outdoors in green spaces.

#### 3.6.1. Maintaining Fitness Levels

I think they (the participants) wanted to maintain the work they’d done so far, the work they’d done through CARE, so keeping that, maintaining that fitness, and improving that again. Then also that social interaction and I think a little bit of boredom and routine so, right I’m going to do two online Zoom sessions a week with CARE and the video that is sent out. They get the real physical benefits of exercise, but it’s more than that now, it’s that social and psychological interaction. (Staff 2)It gets you moving and doing stuff because the thing, two sides. From the counter perspective, exercise is very important to alleviate the issues around chemo. And the chemotherapy does take, knock your body so you can’t do things. And if you don’t do exercise very quickly, I’ve seen people go down a slippery slope of not being able to do anything. And then you get psychologically traumatised in that you can’t do things. It becomes a vicious circle. And I think the exercising also lifts your endorphins. (CARE 6)

#### 3.6.2. Social Support from the CARE Staff

There’s lots of positives. First and foremost were the guys who arrange it, they’re great all of them, they really make you feel welcome, they really help you, it’s worth going along just to see them, plus obviously the exercise benefit. It’s a support. At the end of the sessions they’re [the staff] are always free at times for asking if anyone has any questions or need help or need to know anything. (CARE 9)I mean, we’ve been getting weekly emails from the instructors with all the details of all the sessions and everything on. But it’s the fact that they have kept in touch, and they do so before in a way as wellbecause they would always make sure we knew what was happening when we went to the sessions and keeping that contact, I think, has been very important. (CARE 1)They [the staff] complement one another very well. I mean on Monday, they’re very gentle but firm with the instructions to make sure that you’re doing it properly and not going to injure yourself. But also encouragement and encouraging. (CARE 6)The instructors didn’t just do the exercise programme, they did other things as well. If you didn’t want to do that, there were quizzes. There were various things they’d put on. They were sending out exercise programmes you could look at and follow if you didn’t want to actually do the Zoom programme. So, there were lots of options. So, nobody was left out for whatever reason which is great. A lot of places wouldn’t go to that much trouble. (CARE 2).

#### 3.6.3. Social Support from the CARE Participants

I mean that’s one of the good things about CARE is that, you know, you can talk about it because other people are going through, not the same, I mean I haven’t had anyone with the same cancer. But it’s very similar experiences. (CARE 8)Attending CARE in person gave me someone to speak to who are in a similar situation to myself, I got into to it really, but I stopped going [when] the pandemic started. (CARE 5)But it was just the contact and seeing the faces. That was great. You know, being able to speak to somebody and share, you know, how people were feeling during lockdown as well because it’s not just about the exercise. It’s also the camaraderie and catching up. (CARE 2)What works for me is that I was seeing people who I’d seen before we went into lockdown. So, you know, you sort of make friends with people and you form this little community and we’re such a mishmash of people which is lovely. (CARE 4)

#### 3.6.4. Adapting the CARE Sessions to Meet Participant Needs

Doing session when you’re in your own home, it did remind you how you can use household props and you can adapt what are sort of routine things into your homes. (CARE 8)The CARE sessions are adapted to meet your needs. (CARE 5)

#### 3.6.5. Accessibility of the Online CARE Sessions

For some participants, engaging in CARE online was more convenient as it did not involve travelling to the venue and taking public or shared transport during the COVID-19 pandemic.

I can’t drive now and had to sell my car. And so, it meant I didn’t have to involve my husband usually and lifts and or catch public transport, which is obviously a big boost because I didn’t feel inclined to do that during COVID restrictions. I don’t say this last year has been more difficult, in some ways it’s been easier for me because it’s on Zoom, I don’t have to get out of the house, in some ways that’s better. (CARE 8)You’re not having to travel all the way through the traffic to get to the Portland you can just do it last minute or whatever, so it doesn’t matter if it’s absolutely pouring with rain or whatever. And you can fit it in very easily because it’s just that hour. Whereas when you’re going face to face, you’ve got to allow for travel time so then that will take you all morning to actually get there. Sometimes it’s really busy. So, it’s more convenient [online]. (CARE 4)

#### 3.6.6. Establishing a Routine and Focus

For other participants, scheduling CARE sessions either in person or online helped to create a routine.

I do need the encouragement to get out because with the depression and anxiety, I don’t really get out much so that’s the reason I like to go to the sessions because it gave me an excuse to get out of the house and do something but saying that it is a lot easier to do it on Zoom. (CARE 9)I think, just to keep some sort of focus because I was retired at that time, my life had not come to an end, but it had stopped really. My husband was at work. Nothing had really changed for him. So, going to gym three times a week, it was just that routine. It was keeping in contact with people, having a bit of a laugh and also, helping my physical health as well. (CARE 4)

Participants also reported that CARE was a bona-fide exercise session.

I think the fact that I feel I’m participating in a proper piece of exercise like you might at a gym. Because, you know, you have the warmup with them first of all. Then you do the activities and then you have a cool down which are things that I know you would do if you were doing proper gym work. (CARE 1)

#### 3.6.7. Costs of Engaging CARE Sessions in Person

In some instances, people found the expense of engaging CARE online less costly.

It’s helped me financially because as I said before I found it difficult after six months when you have to start paying because not only was it £5.00 per session, I had to get to Nottingham which costs me another £5.00 to £10.00 as well depending on whether I’m on a train or car, and so in some respects it’s been easier for me this last year. (CARE 9)

#### 3.6.8. Technology Hacks

Participants reported the hacks and techniques they used which helped them to exercise during the CARE sessions.

The PC screen acted as a mirror, where if you’re in a studio you, or a gym, you usually have a mirror. Having yourself in front of you on the screen you could adjust your movements, you know. You could get your arms level…, stand up straighter, look ahead. And I found that was the surprising thing. (CARE 8)

#### 3.6.9. Mental Health/Confidence

It clears your head. Gets rid of headaches. Just makes you feel more with it. More alive. And it stops- I go through bouts of being very restless and fidgeting and that sort of thing and it gets you focused on something. So, yes. And walking helps me switch off from things. (CARE 3).I think if I hadn’t of had that interaction with CARE three times a week for more or less the last year or whatever, I’m not sure whether I would have had the confidence to have gone back. (CARE 4)

### 3.7. Feelings about the Future Participation in CARE Sessions

Participants reported their aspirations for participation in CARE following the easing of COVID-19 restrictions and the opening up of indoor exercise.

I need to do some PA, I am itching to get back to CARE to get back my mobility and flexibility, it will be tough, but there is no pressure, if you want to push yourself you can. (CARE 5)I’d be happy to go in inside now but they’re saying today they’re not going inside for various reasons for a few weeks yet but, yes, I have no qualms about any of that anyway, I’d be happy to all the time. (CARE 9)I’m looking forward to getting back to being inside. I’m looking forward to that. That will be good, to use all the different equipment and I know that’s going to take time or whatever. (CARE 4)

Other participants were more cautious about returning to face-to-face sessions delivered at indoor venues.

It does make you more cautious. And it does make you worry about stuff that you didn’t used to worry about, like going into a café, which is a crazy thing to worry about, but you know. So I would like to go back inside and go back to it. And fit it in around work. But now I, you know, I will continue going to the outdoor sessions because they’re great. (CARE 7)I’ve had my two jabs, I mean I know most- Well I know other people have in the group. And I’m not anxious. But I’m slightly apprehensive, that’s all. So, but I will go back. I mean if moves totally inside then I will go back to it. (CARE 8)

## 4. Discussion

This is one of the first studies to investigate the PA experiences of PLWBC who engaged in a cancer and exercise rehabilitation programme offered by a Football Club Community Trust during the COVID-19 pandemic. National PA agencies have highlighted the need to recognise the importance of PA and PA services during the COVID-19 pandemic [32], both generally, as well as for those people with long-term conditions, including cancer [29]. Indeed, PA has been identified as being an important component of cancer survivorship [17,33] and the pandemic has resulted in a decrease in PA of PLWBC [25]. Recognising the importance of PA for prostate [34], breast [25] and other cancers [35], our research confirmed that the adapted CARE programme was an important ingredient in helping PLWBC to keep physically active during the COVID-19 restrictions. Regardless of whether attendees engaged CARE online or outdoors, sessions provided an important opportunity to secure the physical, social [1,19] and mental health benefits of PA [1,19,36,37]. Moreover, our study identified new opportunities and participant preferences for engaging in PA following the easing of COVID-19 restrictions. Listening and understanding to participant influences, facilitators and barriers to PA has been identified as important in helping people to be active as society emerges from COVID-19 [2]. It also helps to understand the best ways of supporting PLWBC to make and maintain lifestyle changes, including PA [29]. With that in mind, we discuss the barriers and facilitators experienced by CARE participants as well as some of the practical considerations that will be useful for deliverers.

### 4.1. Physical Activity Participation

All the participants had attended the CARE programme pre-pandemic, but the emergence of the COVID-19 restrictions disrupted their engagement in CARE sessions previously provided indoors at the Portland Centre and face-to-face [19]. In some instances, participants reported that the cessation of the face-to-face CARE exercise classes delivered indoors had initially negatively impacted on their PA routines. Research has shown how the government restrictions have had a detrimental impact on people’s routines, including accessing services [23], while other research has indicated that the COVID-19 pandemic has resulted in reductions in PA levels [2], including the PA of breast cancer survivors [25]. In this study, some CARE participants reported that their PA participation had lapsed at the start of the COVID-19 restrictions. However, other attendees also reported adopting self-directed PA such as walking the dog or going to online exercise classes during periods of COVID-19 restrictions, in order to keep active. CARE, including exercise sessions delivered outdoors in green spaces and/or CARE sessions delivered online, were the modes of delivery that participants reported on most frequently, although some reported being aware and using the exercise sheets and the videos that were sent out by the CARE staff.

The organised CARE sessions were an important, if not essential, ingredient of participants scheduled weekly PA. Several attendees reported that the structure of the classes i.e., the warmup, the main session, the cool down and the weekly day and time, were important in providing both routine and organisation to their weekly exercise regimen. Many participants reported how they stood to gain from the well documented benefits associated with regular PA participation [1]. Conversely, some participants who had not engaged with CARE at the start and throughout the periods of government restrictions, reported performing no, or little PA and, in some cases, they felt that their health fitness and functional capacity had deteriorated because of their period of physical inactivity. Occasionally, they reported that declines in their fitness, functional capacity and wellbeing was due to their non-engagement with CARE specifically and that the online and outdoor modes were not preferable. In the case of others, when online sessions became available, CARE sessions were an important moment to arrest lapses in PA participation and take control of their inactivity [19].

This is important given that the government recommendations identify the important role that PA can play not only in managing long-term conditions, but also for developing strength and flexibility, which is important for maintaining functional capacity needed for daily living [1]. Rutherford’s investigations into the CARE programme identified that attendees of face-to-face exercise classes reported improved quality of life and enhanced physical functioning [19], so it is understandable that participants might feel that their non-engagement in CARE sessions, may have negatively impacted on their fitness and functionality. Moreover, we spoke with participants who recognized how easily they could lose their fitness and their loss in functionality could also spiral out of control rapidly and they were keen to guard against this. More positively, routine PA is important to offer pleasant situations at some moment in the day [38] and many participants reported physical, social and mental wellbeing from being active [1] at CARE. Regardless of the mode of participation, CARE sessions were important in maintaining or re-establishing a routine and which participants could use to scaffold other PA and social activity around. Further, although some people did their own PA alongside CARE sessions, several participants referred to the credibility and propriety of attending a bona fide structured CARE exercise session led by qualified instructors. The COVID-19 pandemic and the cessation of CARE sessions at the Portland Centre has resulted in both barriers as well as new opportunities for participants to be physically active. Our study provides insights into the practical considerations of delivering CARE through the different modes of delivery.

### 4.2. CARE Sessions Delivered Outdoors in Green Spaces

Several participants did not like the CARE sessions delivered outdoors. Limitations to the range of exercise equipment that could be transported to the outside venue and made available was reported as significantly reduced, compared to CARE delivered at the Portland Centre. The absence of toilets and changing facilities for people and the unpredictable and inclement UK weather made the delivery outdoors non-preferable to some participants. Similarly, there were exceptionally safety considerations such uneven surfaces where the sessions took place. That said, when it was permitted under COVID-19 guidelines and before sessions were formally delivered outdoors by NCF, several participants felt that organising their own version of CARE outdoors by meeting up with participants and following an exercise regimen was imperative to retain their fitness and PA. It also illustrates how some participants galvanised themselves to keep active the best they could and organising the equipment, hygiene measures, social distancing and routines to help other participants. Indeed, previous research has shown how CARE participants self-organised to engage in external physically active events or self-organise cycling and walking activities [19].

Following the changes to COVID-19 restrictions when PA providers could offer their services outdoors, NCF were able to deliver the CARE sessions in a local green space. Research has highlighted the social benefits of group exercise [11,19], and CARE delivered outside and in person provided an important opportunity for attendees to see each other face-to-face, in real life, albeit it in a socially distanced format. Research also shows the importance of the social aspects of being physically active, which can act as an influential determinant for participation [39]; this is especially important for people who have felt socially disconnected due to COVID-19. CARE outdoors provided attendees with a physical and social connection to likeminded people who knew what it was like to survive cancer and to keep physically active during their survivorship [19]. This social connection was not always experienced in the same way through mainstream exercise provision such as going to a gym [19] or the online CARE delivery. Indeed, some of the practicalities of creating social opportunities online through breakout rooms did not work as well as anticipated. Showing people how these types of functions can be used to facilitate social connectedness and this is a future consideration and something that some participants requested in this study. Participants also reported enjoying the connection with the natural environment, including the nice weather and the seasonal colours associated with the change from Spring to Summer to Autumn seasons which enhanced the participants’ exercise experience. Research has highlighted the role of PA in connecting people to the natural environs and the importance of such connections for wellbeing [40], including benefits to mental wellbeing [41] and this featured in the reports from participants and is a facilitator that can be explored further with participants in meeting their exercise preferences.

We encountered participants who were fearful about contracting COVID-19 through indoor exercise classes [26]. However, because the risk of the transmission of COVID-19 is much reduced outdoors [21], CARE delivered in outdoor venues helped to reduce the anxieties participants held. It is unsurprising that some participants said that attending CARE outdoors was a preferred PA option and this may hold implications for how exercise providers deliver their services going forward in meeting the personalised care needs of participants, including local authorities and charities who provide and facilitate access to green spaces. Further, attending group sessions outdoors may also act as a transition stage for individuals who are worried, but who eventually wish to return to group exercise classes held indoors in the future. Conversations with participants about their needs and future exercise preferences, personalised care and how these can be best met is an important consideration for providers going forward [2].

### 4.3. CARE Online

Research has shown that digital platform users were more likely than non-users to meet PA guidelines during the COVID-19 stay-at-home restrictions in April and May 2020 [42] and CARE sessions online helped participants to keep physically active. Some participants also reported wanting to continue with this digital option as an alternative or complimentary mode of participation in the future and this also reflects participant aspirations for personalised care. The convenience and low costs (i.e., travel, costs, time, and parking) and where mobility issues made this delivery option both acceptable and affordable to some participants. Moreover, for those attendees who expressed difficulties and worries about travelling to venues, including sharing public transport, fears of attending indoor sessions and mixing with other participants in indoor public spaces and the transmission of COVID-19, the online CARE classes provided an appealing alternative to the face-to-face exercise classes held normally indoors. That said, not all our participants found the online CARE sessions to their liking and chose not to engage in this mode of delivery citing several barriers to engagement. Demographic data show that the sample was an older population and that there was variation in the preparedness of some participants to use IT effectively to access online sessions. As such, this is an important consideration when planning online PA delivery. This, in part, reflects ‘a digital divide’, which refers to the inequalities in the access to and availability of infrastructure and skills to effectively engage digital platforms [43].

Staff and participants encountered technical issues with the availability, accessibility and useability of the software applications and IT hardware. The internet speed and loss of broadband connection meant that for some participants, the CARE sessions ‘dropped out’. Furthermore, not all participants were familiar with the procedures of operating the software, the etiquette of attending online events and the general comfort of seeing themselves and others on a screen. Participants also expressed that because of their own functional limitations, occasionally the pace in which the exercise sessions were delivered was sometimes too quick, with more time needed for people to set themselves up to perform the exercises, especially if they were adapting the exercise around their limitations in their fitness and functionality. It reflects the challenge of catering for multiple participants with often complex and diverse needs in an online class situation. It is further complicated when instructors cannot see participants because the participants’ cameras are not turned-on, or participants do not have them. Where participants’ cameras were not available or operable, instructors had to grapple with the challenge of viewing multiple attendees in miniscule display forms on their screens, which might not always offer a good indication of where participants are at in performing an exercise routine. Indeed, and very exceptionally, some of the participants perceived that instructors were not able to watch attendees as thoroughly as they would in person at an indoor class. These are important delivery considerations for offering online classes in CARE and more widely, especially given the focus of online and digital approaches to PA promotion.

Instructors reported the need to create social opportunities for online participants, but participants indicated that this was not the same as sessions delivered in person and logistical challenges were experienced using the breakout rooms for social events. There was also the challenge of creating a safe space for participants to talk with instructors about their PA and health needs, which due to the online format meant privacy was never guaranteed, with some preferring to email instructors for advice and so opportunities for participants to speak with instructors is an important consideration for deliverers. That said, there were participants who persevered with the challenges of the online delivery, got help to deal with the technical issues from family members and came up with some solutions such as increasing their data allowance, purchasing faster broadband, and running their online sessions through a Smart TV screen, to make online exercise delivery work for them. In part, this also reflects both a determination and an adaptability seen by CARE participants to do what they needed to maintain their PA as best they could, as this was essential to their continued wellbeing.

In other considerations, participants reported the physical deconditioning associated with their period of inactivity, combined with the loss of skills, routines, and familiarisation. Following the easing of restrictions and the resumption of face-to-face services, this creates additional challenges for participants and practitioners alike as participants are inducted back to PA programmes [2]. For some people, it will feel like they are starting again, and they will require additional support and help when recommencing their programme of exercise following a period of inactivity [2]. That said, participants in all modes spoke very highly of the efforts that the CARE instructors made to support participants to keep active, making the extra efforts, their ability to adapt exercises to meet functional limitations and to encourage and motivate attendees to be active through these modes of delivery. In short, the staff were praised for being ‘people people’ and in this study we came across staff, employed by a charity who made a concerted effort to get to know and help participants keep active during the COVID-19 pandemic, they also questioned themselves as to what more they could be doing to help people keep active. It reflects the importance of having the right mix of staff to deliver interventions [6,19,44] and efforts to personalise the care offered to PLWBC. Further, our study reports the need to make sure staff are trained and supported to deliver digital health and PA improvement programmes, as participants asked that they be shown some of the basics of how to use online platforms when face-to-face CARE sessions recommenced.

### 4.4. CARE in the Future

Participants had mixed emotions about engaging in CARE through face-to-face delivery at indoor venues, with many constituents’ keen to reengage, while others were more cautious or even reticent [2]. For some, the opportunity to engage online was more convenient, comfortable and affordable. While for those who were fearful of the increased risk of the transmission of COVID-19 within indoor venues, outdoor sessions were seen as a safer option. In part, participants had their horizons widened by these new modes of delivery and some participants hoped these new modes would be offered in the future. Further, online PA provision can provide people an opportunity to keep active during periods of restrictions when people are required to stay at home and or when indoor PA provision is closed due to COVID-19 [42]. Restrictions can also be at the individual level when PLWBC are required to reduce their social contact for instance when they are in receipt of treatment and are immunocompromised. Our paper reflects some of the challenges that deliverers face when providing different modes of provision which meet the needs and aspirations of different groups of participants. Further research into the impact and implementation, including bi-mode provision, where participants can engage PA through a combination of different platforms would be valuable, including those provided through football community foundations. People who elect not to return to face-to-face PA sessions or who have sadly and regrettably lost their lives during the pandemic, will also impact on the social dynamic of organised PA sessions; this was highlighted as an essential ingredient of CARE. As such, this also poses considerations for how deliverers manage these dynamics when programmes recommence. Further, given the emotional component associated with cancer survival, it is important that delivery staff are safeguarded and supported when dealing with the stressful consequences sometimes associated with supporting PLWBC. The outcomes and practical considerations emerging from this study have been shared with the CARE delivery and management teams in order that the learning emerging from this research can help inform intervention design and future delivery plans and as part of the CARE Annual Review. The outcomes have also been shared so they can be disseminated to service commissioners. Further, given some of the concerns participants reported about returning to CARE held face-to-face and indoors, future research should investigate how participants found this transition to ‘normal service’ (i.e., circa pre-pandemic) and the learning that can inform the implementation of CARE going forward. This forms part of future research plans for the research team.

### 4.5. Limitations and Strengths of This Research

There is a need for greater representation from people who did not take part in CARE during the restrictions, to understand their reasons for non-participation. In part, this reflects the challenge of encouraging people to take part in research to explore why they are not engaging in an activity which is inherently positive for their wellbeing [1]. It is plausible to conclude that talking about their non-participation is not always a comfortable experience for non-attendees. In line with the underrepresentation of participants from of Black, Asian and Minority communities in CARE [19], we had no interviewees from BAME groups in this study which is a limitation.

Regarding research strengths, our study provided in-depth and impactful insights into the PA experiences of PLWBC during the COVID-19 pandemic, including their perspectives of engaging the different CARE delivery modes identifying. This is helpful when shaping local programme delivery plans for the future. Moreover, our volunteers offered informative accounts for how they viewed their future engagement in CARE following the easing of COVID-19 restrictions. Further, we sought the views of the staff who delivered the programme, who provided informative perspectives on the delivery of the CARE programme through online and outdoor modes of provision during unprecedented time of healthcare delivery. Our study adopts a bottom-up approach, by sharing participants’ experiences called for in intervention guidance [45] and provides stakeholders of CARE with a voice to share their experiences during the pandemic and called for in the literature [23]. Further, in the discussion, this research also provides important practical considerations for how PA services were offered to PLWBC during periods of government COVID-19 restrictions, including what works well, as well as what does not, which is often overlooked in evaluation reports [46]. This study includes valuable feedback to green space, PA and leisure services. This information is potentially useful for PA providers as they aspire to meet the needs of such groups through personalised care [47]. Regardless of the extant status of COVID-19 restrictions, outcomes of this information are useful, especially when PLWBC need to self-limit their social contact when they may be immunosuppressed or are worried about social contact and risk of COVID-19 transmission. Further, CARE has strong links to the research priorities of the National Cancer Research Institutes, which, in part, aim to establish the best ways to support PLWBC to make lifestyle changes to improve their health [29] through personalised care [47].

## 5. Conclusions

In this study, PLWBC adopted CARE online and outdoors, as well as self-directed activities to maintain their PA levels following the cessation of face-to-face exercise classes held indoors due to COVID-19 restrictions. In turn, these created difficulties and new opportunities to be physically active, each offering barriers and facilitators to participants, which should be considered by deliverers. For CARE online, the barriers included the acceptability and availability of IT hardware, infrastructure and participant skills and comfort. The facilitators included the convenience, comfort, low costs, no travelling and not sharing public transport, as well as maintaining fitness. For CARE outdoors, barriers included the cold weather, lack of toilets and seats and a limited variety of equipment and exercise routines. Facilitators included socialising and meeting up with fellow participants in person, the connection to nature and the natural environment and maintaining fitness levels and mental wellbeing. Further, given the importance of the instructors, it is important to support the staff and leaders who provide these services, including training, equipment and guidance. The CARE programme offered online and face to face at outdoor venues provided an especially important service in helping PLWBC physically active during this period. Collectively, these are important implementation considerations. Given that governments have reinstated further COVID-19 restrictions in the Winter of 2021–2022 and at the time of writing, the unpredictability of the recovery from the pandemic, our study also has discussed the useful practical considerations for meeting the PA needs of this group.

## Figures and Tables

**Table 1 ijerph-19-02945-t001:** The demographic profile of the CARE participants.

Participant	Gender	Age Group (Years)	Cancer	CARE Mode	Self-Organised PA
1	F	55–64	Bowel cancer	No CARE	√
2	F	55–64	Hodgkin’s lymphoma	CARE Online	√
3	F	55–64	Breast cancer	CARE Exercise Videos/Online	⨯
4	F	55–64	Breast cancer	CARE Online	√
5	F	55–64	Bowl cancer	No CARE	⨯
6	M	65–74	Myeloma	CARE Online/Outdoors	√
7	F	45–54	Breast cancer	CARE Exercise Sheet/Online	√
8	F	55–64	Soft tissue sarcoma	CARE Online/Outdoors	√
9	M	55–64	Prostate cancer	CARE Online	√
Staff	Gender	Age Group (years)			
1	M	25–34			
2	M	25–34			

## Data Availability

The data are not publicly available due to ethical restrictions. Please contact the corresponding author for further information.

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
