# Peer review of "An Investigation into the Physical Activity Experiences of People Living with and beyond Cancer during the COVID-19 Pandemic"

_ijerph, 2022, doi:10.3390/ijerph19052945_

Round 1

Reviewer 1 Report

The study investigates the physical activity experiences of people living with and beyond cancer during the COVID-19 pandemic.

The results of this study are in my opinion of relevance to public health research and practice and would fit the scope of the journal. There are, however, minor concerns which should be addressed.

The limitation of the study is the relatively small sample size consisting of 9 CARE participants and 2 CARE delivery staff members. For this reason, these findings cannot be generalized to the broader community based on this study alone.

Please, consider to remove last two sentences from the abstract. Instead, briefly summarize main recommendations for practitioners.

Page 1, lines 30-32: Participants hoped to return to face-to-face 30 CARE sessions delivered indoors following the easing of COVID-19 restrictions. Practical consider ations are provided for future delivery of PA sessions.

I would suggest to include the demographics and participant profile into the Method section.

Table 1. The demographic profile of the CARE participants

Specify strategies to maintain the PA levels of people living with and beyond cancer during the COVID-19 pandemic.

Page 19, lines 842-844: In this study PLWBC adopted a range of strategies to maintain their PA levels following the cessation of face-to-face exercise classes held indoors due to COVID-19 restrictions.

Specify barriers and facilitators to physical activity.

Page 19, lines 844-845. In turn these created difficulties and new opportunities to be physically active, each offering barriers and facilitators to participants.

Please remove the citation from the conclusion.

Page 19, lines 847-850: In doing so, CARE was an important part of the jigsaw helping to meet participant’s physical, practical, emotional, and social needs which aligns with the NHS Long Term Plan for Cancer that identifies that PLWBC should be provided with personalised care and health and wellbeing support and information [47].

Briefly present practical applications of obtained findings with respect to a specific group of people living with and beyond cancer.

Page 19, lines 850-854: Given that governments have re-installed further COVID-19 restrictions in the Winter of 2021-22 and the unpredictability of the recovery from the pandemic, our study also offers useful practical considerations for meeting the PA needs of this group but has broader translation to other groups who wish to maintain their PA.

Author Response

Dear Colleague,

Thank you for reviewing our paper and your comments.  We really do appreciate your time.  We have provided a response to your comments below.

Please, consider to remove last two sentences from the abstract. Instead, briefly summarize main recommendations for practitioners.

Page 1, lines 30-32: Participants hoped to return to face-to-face 30 CARE sessions delivered indoors following the easing of COVID-19 restrictions. Practical considerations are provided for future delivery of PA sessions.

Line 31-32: We have removed this passage 

Line 26-32 We listed the barriers and facilitators which form the main considerations for practicioners.

In line 868-877, we have expanded the conclusion to include these considerations here as well as the discussion

I would suggest to include the demographics and participant profile into the Method section. Table 1. The demographic profile of the CARE participants

Thanks for this, we have considered your suggestion, but as the demographics are findings/results, we have retained these in the results/findings section.  

Specify strategies to maintain the PA levels of people living with and beyond cancer during the COVID-19 pandemic.

Page 19, lines 842-844: In this study PLWBC adopted a range of strategies to maintain their PA levels following the cessation of face-to-face exercise classes held indoors due to COVID-19 restrictions.

Page 19 line 864-866 we have added these strategies:  In this study PLWBC adopted CARE online and outdoors, as well as self-directed activities to maintain their PA levels following the cessation of face-to-face exercise classes held indoors due to COVID-19 restrictions.

Specify barriers and facilitators to physical activity.

Page 19, lines 844-845. In turn these created difficulties and new opportunities to be physically active, each offering barriers and facilitators to participants.

Page 19 (lines 868-874) we have added the following: For CARE online, barriers included the acceptability and availability of IT hardware, infrastructure and IT skills. The facilitators included convenience, low cost, no travelling and maintaining fitness. For CARE outdoors, barriers included cold weather, lack of toilets and limited variety of equipment and exercise routines. Facilitators included socialising and meeting up with fellow participants in person, the connection to nature and maintaining fitness levels

Please remove the citation from the conclusion.

Page 19, lines 847-850: In doing so, CARE was an important part of the jigsaw helping to meet participant’s physical, practical, emotional, and social needs which aligns with the NHS Long Term Plan for Cancer that identifies that PLWBC should be provided with personalised care and health and wellbeing support and information [47].   

We have removed this passage from the conclusion .  

Briefly present practical applications of obtained findings with respect to a specific group of people living with and beyond cancer.

Page 19, lines 850-854: Given that governments have re-installed further COVID-19 restrictions in the Winter of 2021-22 and the unpredictability of the recovery from the pandemic, our study also offers useful practical considerations for meeting the PA needs of this group but has broader translation to other groups who wish to maintain their PA.

In line 868-76 we list the barriers and facilitators plus other considerations and these form the practical considerations that should be borne in mind when meeting the needs of this group of participants.     It now reads:

In turn these created difficulties and new opportunities to be physically active, each offering barriers and facilitators to participants which should be considered by deliverers. For CARE online, the barriers included the acceptability and availability of IT hardware, infrastructure and participant skills and comfort. The facilitators included the convenience, comfort, low costs, no travelling and not sharing public transport and maintaining fitness. For CARE outdoors, barriers included cold weather, lack of toilets and seats and a limited variety of equipment and exercise routines. Facilitators included socialising and meeting up with fellow participants in person, the connection to nature and the natural environment and maintaining fitness levels and mental wellbeing. Further it is important to support the staff and leaders who provide these services including training, equipment and guidance

Thank you for your support in reading our manuscript.

best wishes

Andy Pringle

Reviewer 2 Report

This research provides important practical considerations for how PA services are offered to PLWBC during periods of government COVID-19.

Even if the number of subjects is small and an objective statistic cannot be achieved, the study may be very useful for practitioners.
I suggest you make minor revisions:

  • Please add the results in the abstract and better score the conclusions of your study;
  • You may be more explicit in your conclusions in relation to the aim of the paper.

Author Response

Dear Colleague,

Thank you for reviewing our paper, we appreciate your time.   Regarding the specific comments:

Even if the number of subjects is small and an objective statistic cannot be achieved, the study may be very useful for practitioners.   

Thank you, we agree the paper has some useful practical considerations.

Please add the results in the abstract and better score the conclusions of your study; 

Page 1 Abstract:

Thanks for this suggestion in the abstract we already reported the key findings, but we have amended the abstract to make it clear these are findings   " The findings show that the COVID-19 pandemic and government restrictions impacted on PA participation, yet exercise sessions provided via CARE offered participants an important opportunity to arrest their inactivity, keep active and maintain their fitness and functionality. Barriers to participation of CARE online included access to IT infrastructure, internet connectivity and IT skills. Regarding CARE outdoors, the weather, the range of equipment and lack of toilets were barriers. In the different CARE modes, the skills of programme delivery staff who were sensitive to the needs of participants, social support, and the need for participants to maintain good mental and social health were important facilitators for engagement.

Line 32 we have also scored the conclusion which now reads:

Line 32: CARE helped PLWBC to keep physically active. 

You may be more explicit in your conclusions in relation to the aim of the paper.

Lines: 864-887:  We have re-focussed the conclusion which now reads: 

In this study PLWBC adopted CARE online and outdoors, as well as self-directed activities to maintain their PA levels following the cessation of face-to-face exercise classes held indoors due to COVID-19 restrictions In turn these created difficulties and new opportunities to be physically active, each offering barriers and facilitators to participants which should be considered by deliverers. For CARE online, the barriers included the acceptability and availability of IT hardware, infrastructure and participant skills and comfort. The facilitators included the convenience, comfort, low costs, no travelling and not sharing public transport and maintaining fitness. For CARE outdoors, barriers included cold weather, lack of toilets and seats and a limited variety of equipment and exercise routines. Facilitators included socialising and meeting up with fellow participants in person, the connection to nature and the natural environment and maintaining fitness levels and mental wellbeing. Further given the importance of the instructors, it is important to support the staff and leaders who provide these services including training, equipment and guidance.       The CARE programme offered online and face to face at outdoor venues provided an especially important service in keeping PLWBC physically active during this period. Collectively these are important implementation considerations . Given that governments have re-installed further COVID-19 restrictions in the Winter of 2021-22 and at the time of writing, the unpredictability of the recovery from the pandemic, our study also has discussed the useful practical considerations for meeting the PA needs of this group..

Thank you once more for reviewing our paper.

Best wishes

Andy

Reviewer 3 Report

This is a well written manuscript addressing a public health need using an interesting method (football clubs) in its promotion.

Low 'n' and lack of diversity among participants greatly limits study (although this is mentioned in limitations)

Manuscript a bit lengthy

Addressing next steps in a more detailed fashion would add value 

Author Response

Dear Colleague,

Thank you for your time reviewing out paper, we appreciate your efforts.     

This is a well written manuscript addressing a public health need using an interesting method (football clubs) in its promotion. 

Thank you, we also agree and feel the paper will be of interest to the readership given the unique topic and setting.

Regarding your specific comments:

  • Low 'n' and lack of diversity among participants greatly limits study (although this is mentioned in limitations).  We agree and as you indicate we have reported this in the limitations.  However a strength is the depth of content from the interviews on a topic that has not up until now been investigated with this group and in this setting.   As you say, it is interesting.
  •  
  • Manuscript a bit lengthy.  The paper is a bit longer, this in order to capture the depth of participant and staff experiences engaging the different modes of CARE (e.g. online and outdoors), we have had to allocate the necessary space to include this depth. However, we have tried to be concise in the discussion and make the conclusion more focussed.  Also reviewers 1 and 3 have requested some additional content be added to the manuscript, so this has slightly extended the manuscript.  We hope our explanation is to your satisfaction. 
  •  
  • Addressing next steps in a more detailed fashion would add value: Thank you once more for a helpful comment, we have added some content regarding next steps.
  •  
  • Line 822-830: we have added: 

    The outcomes and practical considerations emerging from this study have been shared with the CARE delivery and management teams in order that the learning emerging from this research can help inform intervention design and future delivery plans. The outcomes have also been shared so they can be disseminated to service commissioners. Further, given some of the concerns participants reported about returning to CARE held face-to-face and indoors, future research should investigate how participants found this transition to ‘normal service’ (i.e., circa pre-pandemic) and the learning that can inform the implementation of CARE going forward. This forms part of future plans for the research team     

  •  
  • Best regards.   Andy Pringle